# In Vitro Microevolution and Co-Selection Assessment of Amoxicillin and Cefotaxime Impact on *Escherichia coli* Resistance Development

**DOI:** 10.3390/antibiotics13030247

**Published:** 2024-03-07

**Authors:** Ádám Kerek, Bence Török, Levente Laczkó, Zoltán Somogyi, Gábor Kardos, Krisztián Bányai, Eszter Kaszab, Krisztina Bali, Ákos Jerzsele

**Affiliations:** 1Department of Pharmacology and Toxicology, University of Veterinary Medicine Budapest, H-1078 Budapest, Hungary; bencetorok99@gmail.com (B.T.); somogyi.zoltan@univet.hu (Z.S.); banyai.krisztian@univet.hu (K.B.); jerzsele.akos@univet.hu (Á.J.); 2National Laboratory of Infectious Animal Diseases, Antimicrobial Resistance, Veterinary Public Health and Food Chain Safety, University of Veterinary Medicine Budapest, H-1078 Budapest, Hungary; kg@med.unideb.hu (G.K.); kaszab.eszter@univet.hu (E.K.); bali.krisztina@univet.hu (K.B.); 3One Health Institute, University of Debrecen, Nagyerdei krt. 98, H-4032 Debrecen, Hungary; 4HUN-REN–UD Conservation Biology Research Group, Egyetem tér 1, H-4032 Debrecen, Hungary; 5National Public Health Center, Albert Flórián út 2-6, H-1097 Budapest, Hungary; 6Department of Gerontology, Faculty of Health Sciences, University of Debrecen, Sóstói út 2-4, H-4400 Nyíregyháza, Hungary; 7Veterinary Medical Research Institute, H-1143 Budapest, Hungary; 8Department of Microbiology and Infectious Diseases, University of Veterinary Medicine, István u 2, H-1078 Budapest, Hungary

**Keywords:** microevolution, co-selection, MEGA-plate, *Escherichia coli*, amoxicillin, cefotaxime, NGS

## Abstract

The global spread of antimicrobial resistance has become a prominent issue in both veterinary and public health in the 21st century. The extensive use of amoxicillin, a beta-lactam antibiotic, and consequent resistance development are particularly alarming in food-producing animals, with a focus on the swine and poultry sectors. Another beta-lactam, cefotaxime, is widely utilized in human medicine, where the escalating resistance to third- and fourth-generation cephalosporins is a major concern. The aim of this study was to simulate the development of phenotypic and genotypic resistance to beta-lactam antibiotics, focusing on amoxicillin and cefotaxime. The investigation of the minimal inhibitory concentrations (MIC) of antibiotics was performed at 1×, 10×, 100×, and 1000× concentrations using the modified microbial evolution and growth arena (MEGA-plate) method. Our results indicate that amoxicillin significantly increased the MIC values of several tested antibiotics, except for oxytetracycline and florfenicol. In the case of cefotaxime, this increase was observed in all classes. A total of 44 antimicrobial resistance genes were identified in all samples. Chromosomal point mutations, particularly concerning cefotaxime, revealed numerous complex mutations, deletions, insertions, and single nucleotide polymorphisms (SNPs) that were not experienced in the case of amoxicillin. The findings suggest that, regarding amoxicillin, the point mutation of the *acrB* gene could explain the observed MIC value increases due to the heightened activity of the *acrAB-tolC* efflux pump system. However, under the influence of cefotaxime, more intricate processes occurred, including complex amino acid substitutions in the *ampC* gene promoter region, increased enzyme production induced by amino acid substitutions and SNPs, as well as mutations in the *acrR* and *robA* repressor genes that heightened the activity of the *acrAB-tolC* efflux pump system. These changes may contribute to the significant MIC increases observed for all tested antibiotics. The results underscore the importance of understanding cross-resistance development between individual drugs when choosing clinical alternative drugs. The point mutations in the *mdtB* and *emrR* genes may also contribute to the increased activity of the *mdtABC-tolC* and *emrAB-tolC* pump systems against all tested antibiotics. The exceptionally high mutation rate induced by cephalosporins justifies further investigations to clarify the exact mechanism behind.

## 1. Introduction

Antimicrobial resistance stands as one of the most pressing issues in current animal and public health, causing economic losses in the order of trillions of dollars [1]. The associated human mortality, projected to reach 10 million by 2050, highlights the severity of the problem [2]. *Escherichia coli* (*E. coli*), globally recognized as one of the most prevalent Gram-negative pathogens, holds significant relevance for food-producing animals, pets, and humans alike [3]. It is considered an excellent indicator for antimicrobial resistance [4,5], with a pivotal role in public health as a carrier of resistance genes via the food chain [6]. Resistance of *E. coli* isolates of animal origin to the active ingredient amoxicillin is widely reported worldwide [7,8,9,10], partly attributed to the frequent use of penicillins in veterinary medicine in Europe [11]. In the poultry industry, the use of broad-spectrum cephalosporins has led to the rapid spread of resistance, primarily associated with the dissemination of extended-spectrum beta-lactamases (ESBL) and plasmid-mediated *ampC* genes [12,13]. As a result, the use of cephalosporins in the poultry industry has been banned worldwide [14]. The frequent contamination of commercially available poultry meat can lead to foodborne infections [15,16]. Resistance to broad-spectrum cephalosporins in *E. coli* is known to have a clonal nature, with the ST131 sequence type currently being the most widespread globally [17,18,19,20]. Among the third and fourth-generation cephalosporins, cefovecin is utilized in companion animals [21], while ceftiofur is widely used in food-producing animals, which was permitted for the treatment of day-old chicks and turkeys until the early 2000s. However, this contributed to the rapid spread of resistance against cephalosporins [22].

It is widely accepted that a direct consequence of antibiotic use is the development and spread of resistance, a process induced by mutational events through Darwinian evolution, providing bacteria with an evolutionary advantage [23]. Resistant mutant strains arise stochastically, meaning that resistant mutations randomly appear in the population and then spread under the selective pressure of therapy [24]. However, resistant strains may have a disadvantage compared to unexposed and non-resistant strains, and various resistance mechanisms can interact with each other, triggering non-additive fitness effects. This implies that the combination of different resistance mechanisms may result in different outcomes than their individual presence [25]. These interactions lead to the formation of so-called rugged fitness landscapes, evolutionary maps where some genetic changes are more or less accessible in the population, influencing over time the path a strain follows in terms of adaptation and change [26,27,28]. Identifying these evolutionary compromises forms the basis for combating drug resistance [29,30,31].

During in vitro evolutionary studies, compounds exhibiting collateral sensitivity can be identified by exposing the tested population to increasing concentrations of a given agent, thereby inducing resistance [29,32,33]. The impact of differentiated collateral responses is not fully understood to this day; collateral sensitivity is never universal and, in fact, is rare. Not every mutation or evolutionary trajectory results in collateral sensitivity [34]. The Microbial Evolution and Growth Arena (MEGA-plate), a system developed by Harvard University for evolutionary and co-selection studies [35], has been successfully adjusted and adapted in our methodology [36], and was previously utilized in the investigation of florfenicol resistance [37], which is a giant Petri-dish to growth bacteria. The advantage of this system lies in its ability to subject bacteria to continuous selection pressure, facilitating the stochastic generation of resistant mutations [24].

The aim of this research is to elucidate the evolutionary processes induced by amoxicillin, a widely used antibiotic in poultry, and cefotaxime, a third-generation cephalosporin crucial for public health. This investigation employs in vitro methods to explore the relationships between phenotypic and genotypic changes occurring under the influence of mutational selection pressure. Furthermore, our goal is to uncover explanations for the widespread dissemination of resistance to cephalosporins in poultry, despite its non-authorized nature in Europe.

## 2. Results

### 2.1. Minimum Inhibitory Concentration (MIC) Shifts

In the case of amoxicillin, bacteria started to grow on agar containing 1000× of the antibiotic within 5 days; for cefotaxime, this process required 13 days. Regarding amoxicillin (Table 1), it can be observed that against most antibiotic classes, a significant increase in MIC values occurred, except for oxytetracycline and florfenicol. For cefotaxime (Table 2), this effect was observed for all tested substances.

### 2.2. Extended-Spectrum Beta-Lactamase (ESBL) Production Screening

Table 3 and Table 4 show the results of the test assessing extended-spectrum beta-lactamase (ESBL) production and do not show at least a three-fold MIC reduction in the presence of clavulanic acid for both pairs of substances.

### 2.3. Sequencing Data Quality

The quality of the contigs generated during the genome assembly process was assessed using QUAST (v5.2.) [38]. The results of this quality control analysis are presented in Appendix A.

To comprehensively assess the overall genomic characteristics of each sample, GenomeScope profiles were generated. These profiles provide valuable insights into genome evolution and serve as a foundation for determining parameters for subsequent analyses. The kmer linear plot (Appendix A) following error correction facilitates estimation of coverage, genome size, and kmer size. These qualitative data corroborate the adequacy of sequencing quality and the suitability of contigs for bioinformatic analysis. The kmer frequency histograms exhibit consistent alignment with the expected patterns for *E. coli*. Confirmatory analyses using Checkm (v1.1.6) and Kraken Software (v1.1.1) across all samples yielded a 100% match with the *E. coli* bacterial species.

Genomic diversity analysis among between 0× and 1000× amoxicillin (Appendix A) and 0× and 1000× cefotaxime (Appendix A) genomes was conducted using Average Nucleotide Identity (ANI) v2.0 software. The ANI calculator estimates the average nucleotide identity using the best one-way matches (one-way ANI) and reciprocal best matches (two-way ANI) between two genomic datasets. ANI values among genomes of the same species are typically above 95% [39].

### 2.4. Antimicrobial Resistance Gene (ARG) Set

With regard to the antimicrobial resistance gene (ARG) set, our identified ARGs met the stringent threshold criteria set forth by the CARD database, exhibiting coverage and sequence identity percentages exceeding 90%. These ARGs were consistently detected across all samples, totaling 44 distinct ARGs with the potential to confer resistance to a range of 22 antibiotics, disinfectants, and various dyes. Notably, the presence of *ampC* and *ampH* genes, responsible for imparting resistance to beta-lactam (penam and cephalosporin) antibiotics through enzymatic inactivation, was observed (Appendix A).

For amoxicillin, the *bacA* gene was identified as a mobile genetic element (MGE) in all samples and was traceable on the phylogenetic tree. Additionally, the *emrB* gene was found on the tree in the 10×, 100×, and 1000× samples. In the case of cefotaxime, the *bacA* gene was identified as an MGE, and the *ugd* gene was also identified. The former was present in the 1×, 10×, 100×, and 1000× samples, while the latter was traceable on the tree in the 100× and 1000× samples. Notably, in the cefotaxime 100× sample, the *ampH* gene was found on a plasmid. Various identified multidrug efflux pump genes could potentially play a role in the development of resistance against penams and cephalosporins.

We identified a total of 44 antimicrobial resistance genes (ARGs), categorized by drug group and resistance mechanism (Appendix A). An analysis using mlplasmid v2.1 software revealed that all identified ARGs were encoded within the bacterial chromosomes, with the exception of the *ampH* gene found in the cefotaxime 100× sample. Further investigation using VirSorter v2.2.2 software classified *bacA* and *emrB* as phage-encoded genes. The MobileElementFinder v1.0.3 software also supported this conclusion (Appendix A).

### 2.5. Serotyping and Virulence Factors

The sequencing data provided the basis for identifying the serotype of the examined strain. This led to the discovery of specific polysaccharides for the O6 serotype (*wzx*, *wzy*) and protein-based antigens H1 and H12 (*fliC*). To assess the potential effect of varying concentrations of amoxicillin and cefotaxime on the number of virulence factors, 40 identical virulence factors were consistently observed across all samples. Overall, this suggests that the active substances amoxicillin and cefotaxime did not have a significant impact on the presence of these virulence factors.

### 2.6. Mutations

A thorough analysis of the amoxicillin and cefotaxime samples revealed the presence of a total of 8747 and 9673 mutations, respectively. Of these mutations, 4618 and 4922 had clear functional assignments. The distribution of total mutations varied between 1706 and 2035 mutations per sample, while the distribution of identified mutations ranged from 912 to 1084 mutations per sample. Upon comparison with the baseline 0× sample, the overlap of the identified mutations with the amoxicillin samples was found to be 99.1% (1×), 98.2% (10×), 98.6% (100×), and 100.0% (1000×). Similarly, the overlap of identified mutations with the cefotaxime samples was found to be 101.3% (1×), 101.4% (10×), 118.9% (100×), and 118.3% (1000×). The distribution of each mutation type in the samples is summarized in Table 5 (amoxicillin) and Table 6 (cefotaxime). All mutations are interpreted in relation to the SYNB8802 strain, which served as the reference strain for bioinformatic analysis.

The majority of mutations detected in the samples belonged to the single-nucleotide polymorphism (SNP) category, with the highest frequency observed in the 1000× cefotaxime sample. Subsequent to SNP mutations, the next most prevalent mutations were intricate mutations entailing complete amino acid substitutions, notably present in the 1× sample. Deletion mutations, resulting in the loss of a singular amino acid, were most frequently noted in the 100× and 1000× cefotaxime samples. Compared to the amoxicillin samples, numerous insertions were observed in the 100× and 1000× samples of strains treated with cefotaxime. (Appendix A).

When investigating mutations relevant to antimicrobial resistance, genomic alterations attributed to SNPs were identified, potentially elucidating the elevated MIC values against several antibiotics after exposure to amoxicillin, as detailed in Table 6, while the impact of cefotaxime treatment is reflected in Table 7, Table 8, Table 9, Table 10 and Table 11.

Upon exposure to amoxicillin, a complex amino acid substitution mutation occurred in the *vgrG* gene at the highest drug concentration, the expression of which plays a role in virulence by toxin secretion [40]. No deletion was observed; similarly, an insertion occurred in the *ftsK* gene, which is involved in chromosome segregation [41]. Among the SNPs, a notable nucleic acid variation was observed in the *ampC* gene at the 100× and 1000× concentrations (Table 7), which could explain the significant increase in MIC values against the substances listed in Table 1. In addition, it is essential to highlight the mutation observed in the *acrB* gene, which may lead to increased functionality of the *acrAB-tolC* efflux pump system.

Upon exposure to cefotaxime, significantly more complex mutational changes occurred compared to amoxicillin. The most crucial mutation involving complex amino acid substitutions took place in the promoter region of the *ampC* gene [42] inducing alterations that predict the phenotypic expression of enzymatic inactivation against beta-lactam antibiotics (Table 8).

Exposure to cefotaxime at concentrations of 100× and 1000× resulted in the observation of numerous deletions, primarily affecting genes encoding mutation repair proteins and genes responsible for sensing changes in cell membrane pressure (Table 9).

Several insertions were also observed, with particular significance attributed to the mutation in the *marR* gene (Table 10). This gene functions as a repressor operon, contributing to the development of resistance against various antibiotics such as penicillins, cephalosporins, tetracyclines, rifampicin, chloramphenicol, and fluoroquinolones [43,44].

The most complex changes due to SNPs were observed, with approximately 234 genes undergoing mutation, in response to 10×, 100×, and 1000× concentrations of the active substance compared to the untreated strain. Among these, mutations associated with antibiotic resistance were highlighted in Table 11. It is noteworthy to mention the mutation affecting the functioning of the *acrAB-tolC* pump system, as observed with amoxicillin, occurring in the *acrR* repressor gene [45] and mutations in the *acrB* gene [46,47]. The *robA* gene functions as a positive regulator for genes that encode the *acrAB* efflux pump [48]. Also notable is the variation in the promoter region of the *ampC* gene [42] at concentrations of 100× and 1000×. Mutations were observed in the *mrxA* gene, which plays a role in bacterial cell wall synthesis [49,50], as well as in the *mdtB* gene, a component of the *MdtABC-TolC* efflux complex system [51] and the *emrR* gene, a component of the *EmrAB-TolC* efflux complex system [52].

Regarding amoxicillin treatment, mutations in the *ampC* gene and *acrB* gene could contribute to both the enzymatic inactivation and the enhanced functionality of the *acrAB-tolC* pump system (Appendix A).

Regarding cefotaxime, we observed significantly more complex processes. Among these, we first need to highlight the multiple complex amino acid substitutions occurring in the *ampC* gene promoter region and an undefined mutation, which, based on the analyses, predicted the appearance of phenotypic resistance to various beta-lactam agents. Mutations in the *acrB* gene, as well as mutations in the regulating *acrR* and *robA* genes, determine the operation of the *acrAB-tolC* pump system. The *mdtB* gene encodes a transporter that combines with the *mdtC* gene to form a heteromultimeric complex, thereby constituting a multidrug transporter. This *mdtBC* complex functions as a component of the *mdtABC-tolC* efflux system [53]. In *E. coli*, the *emrR* gene serves as a negative regulator within the *emrAB-tolC* multidrug efflux pump complex [54]. Mutations within this gene lead to the expression of the *emrAB-tolC* complex. Primarily, the *emrA* and *emrB* genes are accountable for expelling enrofloxacin, while the *tolC* gene encodes the efflux pump responsible for eliminating less potent drugs. This phenomenon potentially elucidates the rise in MIC values for other drugs following exposure to high concentrations of cefotaxime. The *acrB* gene constitutes a vital element of the *acrAB-TolC* multidrug efflux complex protein. *AcrB* operates as a heterotrimer, encompassing the inner membrane component, and plays a pivotal role in substrate recognition and energy transduction by acting as a drug-proton antiporter [46,47,55,56]. Mutations induced in the *acrB* gene by 10× and 100× concentrations of cefotaxime contribute to elevated MIC values. As a repressor, the *acrR* gene modulates the activity of the *acrAB-tolC* multidrug efflux complex. Mutations in *acrR* can lead to high levels of antibiotic resistance, as this gene serves as the repressor in the regulation of the *acrAB-tolC* multidrug efflux complex. As a result of mutations, the operation of the *acrAB-tolC* complex may intensify, enhancing resistance to antibiotics (Appendix A).

## 3. Discussion

The practical significance of exploring cross-resistances is crucial, as it can provide important guidance for ruling out second-line therapeutic agents that are likely to be cross-resistant. Nichol et al. found that in the case of cefotaxime resistance, piperacillin or gentamicin proved to be a good alternative. However, for ciprofloxacin, the mathematical probability of cross-resistance was much higher. Notably, significant cross-resistance is observed with fosfomycin, ampicillin, ticarcillin, and cephalosporins [34]. In our investigations, taking these findings into consideration, based on the phenotypic assay results of sensitivity, we found that in the event of amoxicillin resistance, oxytetracycline, florfenicol, and enrofloxacin could be favorable choices based on the extent of the MIC value increase. Cross-resistance may emerge among groups of active substances, particularly those sharing similar or identical mechanisms of action. An established instance is the cross-resistance observed between macrolides and lincosamides [57]. In the case of amoxicillin, it was evident that considerably fewer point mutations occurred, possibly attributed to the absence of cross-resistance induction by the protein synthesis inhibitors oxytetracycline and florfenicol. Conversely, with cefotaxime, the substantially higher frequency of point mutations observed led to alterations that have already elevated the MIC of these antibiotics. It is worth noting that while the role of antibiotic concentrations below the MIC in the development of resistance is often emphasized [58], our results unequivocally demonstrate that high concentrations may also contribute to this process.

The results of evolutionary and co-selection studies support the use of systems such as morbidostat [59] or the MEGA-plate method [35,36], which eliminate the need for repeated bacterial population reseeding, thereby enabling less stochastic population dynamics [34]. However, it has also been demonstrated that sensitivity changes arising from phenotypic plasticity may be reversible within a short period of time [60].

The investigations according to the CLSI protocols for the phenotypic detection of ESBL production yielded negative results, a finding that was further confirmed genotypically, as the genes responsible for ESBL production were not present in the samples. To assess this, sensitivity testing was conducted using cefpodoxime (expressing *TEM*, *SHV*, and *CTX-M* gene hydrolysis), cefotaxime, ceftazidime, ceftriaxone, or aztreonam agents and their clavulanic acid combinations [61], However, it is crucial to highlight that the weak sensitivity of these tests can be attributed to the production of beta-lactamases not inhibited by clavulanic acid, such as *ampC* beta-lactamase or metallo-beta-lactamases [62]. In our metagenomic investigations, we identified the presence of the *ampC* gene in all samples. To mitigate potential false-negative results in this regard, cefepime, which is a weak substrate for most *ampC* beta-lactamases, can be employed. Other strategies include the use of chromogenic agars, agar containing cloxacillin, or the addition of EDTA for metallo-beta-lactamase inactivation [61].

Resistant genetic lines to the 1000× concentration of amoxicillin were selected within only five days, in contrast with the 13 days observed in the case of cefotaxime. However, with amoxicillin, we noted significantly fewer mutations compared to cefotaxime. This could be attributed to differences in contact time, but the higher mutagenic effect induced by cephalosporins is also a plausible hypothesis. Based on our results, it can be observed that even a single dose of amoxicillin induced an increase in the MIC of cephalosporin agents, and higher concentrations further amplified this increase in MIC values. The significance of this finding may contribute to the observed spread of cephalosporin resistance in the poultry industry.

In the case of *E. coli*, the so-called SOS system repairs nearly 30 different genes, encoding proteins involved in DNA repair tasks [63]. Bacterial mutation rates increase due to DNA damage or replication stalling, which is attributed to approximately 30 different genes repaired by the SOS system [64], This explains the observed increase in mutation rates in environments exposed to elevated concentrations of antibiotic agents. These SOS mechanisms ensure the long-term survival of *E. coli*, providing a form of evolutionary fitness [65]. Previous studies have demonstrated the inducing effect of the SOS system in the case of fluoroquinolone agents [66] and beta-lactam antibiotics [67] thereby increasing their mutagenic impact.

Decades of antibiotic use have had a significant effect on the selection and spread of resistant bacteria, and these changes were eventually fixed in the population [68]. However, some studies suggest that bacteria are not passive participants in these evolutionary processes [69]. Antibiotic stress leads to an increased mutation rate (hypermutators), resulting in selection [70]. According to our findings, the results of cefotaxime testing indicate that cephalosporins may act as hypermutators. In comparison to the untreated strain, the 100× concentration of the agent induced 161 SNPs, and the 1000× concentration induced 174 SNPs, contrasting with amoxicillin, which induced only 6 and 5 SNPs, respectively. Furthermore, numerous complex mutations, deletions, and insertions were observed with cefotaxime at these concentrations, which were not observed with amoxicillin. Similar observations were not made in our earlier investigations with florfenicol [37]. The significance of point mutations and recombination [71] is much greater than that of de novo mutations [72]. It is crucial to consider that antibiotic concentrations in the organism show spatial and temporal variations [73], and their concentration gradients may impact the recombination of the commensal microbiome [74].

In *E. coli*, the tripartite resistance-nodulation-division (RND) transporter family member *acrAB-tolC* efflux pump exhibits specificity against several clinically important antimicrobial compounds, with its operation being typically repressed by the *acrR* repressor gene [75,76]. Upon exposure to amoxicillin, we observed an SNP in the *acrB* gene, which could have led to increased functionality of the *acrAB-tolC* efflux pump, explaining the elevated MIC values for the specific agents. The mechanisms behind the observed MIC increases with cefotaxime might involve more complex processes. On the one hand, there could be a complex, multiple amino acid exchange mutation in the promoter region of the *ampC* gene, which provides resistance to beta-lactam agents through enzymatic means. On the other hand, an SNP affecting the *ampC* gene might contribute to these MIC value increases. Additionally, mutations in the *acrR* and *robA* repressor genes involved in the regulation of the *acrAB-tolC* efflux pump system, as well as an SNP in the *mdtB* gene and *emrR* gene, could have contributed to the increased functionality of the *mdtABC-tolC* and *emrAB-tolC* pump systems. Collectively, these changes explain the significant MIC value increases observed against the tested agents. Overall, these multidrug efflux pumps contribute to the increase in the MICs of the active substances tested alongside the treatments [77]. This has been observed, for example, in the cases of colistin, florfenicol, and potential sulfonamides. Our results have practical implications, as they support the conclusion that amoxicillin, which is widely used in practice, contributes to the emergence of resistance to cephalosporins. This resistance can be induced by the presence of even a single concentration of amoxicillin. The clinical significance of the *ampC* gene is that its expression can be induced or is constitutive [78], although it does not generally lead to resistance to fourth-generation cephalosporins or carbapenems, unless its overexpression is associated with a decrease in porin channels [79], which combination of resistance mechanisms may explain the high resistance [80].

## 4. Materials and Methods

### 4.1. Tested Bacterial Strain

The *E. coli* strain utilized in our experiments is the reference strain ATCC 25922 (LGC Ltd., Teddington, UK), which was initially isolated in Seattle in 1946. The choice of this strain was deliberate, considering its widespread use and comprehensive documentation in the scientific literature. This particular strain is highly suitable for resistance studies due to its sensitivity to various antibiotics, enabling a more accurate assessment of the drug’s effects.

### 4.2. Preparation of the MEGA-Plate

The experiments were conducted in a 60 cm × 30 cm polycarbonate tray constructed from 5 mm thick material (Innoterm Ltd., Budapest, Hungary) and bonded together using waterproof tetrahydrofuran adhesive. To facilitate the separation of media with varying antibiotic concentrations in the lower layer, the bottom of the tray was divided into nine equal compartments. Prior to usage, the tray underwent disinfection by filling it with a 7.5% hydrogen peroxide solution (VWR International Kft., Debrecen, Hungary) [57,58], followed by wiping the inner surface and rim of the cover plate with a 1% sodium hypochlorite aqueous solution (Merck KGaA, Darmstadt, Germany). Subsequently, it was placed in a sterile chamber for 15 min, after which the hydrogen peroxide was removed using a vacuum pump and a 30 min UV light treatment was administered. 

During the medium infusion process, three separate layers were established. The bottom layer consisted of nine individual compartments containing antibiotic concentrations ranging from 0×, 1×, 10×, 100× to 1000× of the active ingredient. These concentrations advanced from the tray’s edges inward, specifically within compartments 1–5. The second layer formed a cohesive, continuous solid sheet, promoting uniformity between layers 1 and 3. Layer 3 comprised a semi-fluid medium conducive to bacterial diffusion and growth. Within this layer, bacterial proliferation took place amidst ascending drug concentration gradients.

In preparing the culture medium, we opted for BD Bacto Agar (VWR International Ltd., Debrecen, Hungary) at a concentration of 2%, except for the third layer, where a semi-fluid composition was devised at a concentration of 0.28%. To supplement nutrients, one LB-Lennox capsule (VWR International Ltd., Debrecen, Hungary) was included per liter of medium. In our initial trials, bacterial growth failed to manifest inward from the initial concentration line. To counteract this, an extra capsule was introduced into the upper layer, resulting in the anticipated inward growth pattern. Drawing from our experience, an additional capsule was also included in the upper layer. Cycloheximide (Merck KGaA, Darmstadt, Germany) was introduced into the medium at a concentration of 64 µg/mL to mitigate fungal contamination.

In the lower and middle tiers, a solution of 4 mL per liter of black acrylic stain (Artmie, Budapest, Hungary) was introduced to enhance visibility. One day prior to the commencement of the experiment, a yellow loop containing the *E. coli* strain, preserved in a Microbank system (VWR International Ltd., Debrecen, Hungary) at −80 °C, was transferred into tryptone soy broth (VWR International Ltd., Debrecen, Hungary) and subjected to a 37 °C incubation period lasting 24 h. Subsequently, the *E. coli* strain was inoculated along both edges of the plate, which was then positioned within a 37 °C incubator environment (Figure 1 and Figure 2).

### 4.3. Antibiotic Susceptibility Testing

To begin the experiment, we established the minimum inhibitory concentrations (MICs) of specific antibiotics for the *E. coli* strain. Our investigation included assessing susceptibility to ceftriaxone, cefquinome, cefotaxime, ceftiofur, colistin, enrofloxacin, amoxicillin, neomycin, oxytetracycline, florfenicol, and potentiated sulfonamide. All antibiotics were procured from Merck KGaA (Darmstadt, Germany). We conducted the assay following the methodology outlined by the Clinical Laboratory Standards Institute (CLSI) [81]. To confer an evolutionary advantage for the selection of resistant lines, we designated 0.25× of the initial MIC value of amoxicillin (4 µg/mL) and cefotaxime (0.03 µg/mL) as the 1× concentration. The rationale behind selecting an MIC of 0.25× was to provide bacteria with an evolutionary advantage in the development of resistance. This allowed for the selection of resistant strains through initial sub-dosing. Utilizing concentrations of 10×, 100×, and 1000× relative to this 1/4 concentration, we prepared the MEGA-plate. Samples were extracted from each compartment containing antibiotic concentrations and inoculated onto differentiating and selective ChromoBio^®^ Coliform agar (Biolab Zrt., Budapest, Hungary) to verify the absence of contamination. Our tests were conducted in triplicate, consistently revealing closely aligned trends. Bold values within the table denote an escalation in the MIC (µg/mL) compared to the baseline (Table 1). Bacterial colonies, derived from a colony-forming unit, were transferred onto tryptone-soy agar (Biolab Zrt., Budapest, Hungary) and stored within a Microbank™ system (Pro-Lab Diagnostics, Richmond Hill, ON, Canada) at −80 °C for subsequent use. Each case involved the collection of three replicate samples, with each assay being performed in triplicate. The working plates were then subjected to an incubation period at 37 °C for 18–24 h, during which the MIC values were evaluated through visual observation relative to the positive controls.

### 4.4. Assessment for ESBL Production

The assay for ESBL production adhered to the CLSI methodology [81]. In this analysis, we determined the MIC of the bacterial strains against ceftazidime, ceftazidime-clavulanic acid, cefotaxime, and cefotaxime-clavulanic acid. For each dilution involving clavulanic acid combinations, a consistent concentration of 4 µg/mL clavulanic acid was maintained. Subsequently, the plates underwent incubation in a thermostat set at 37 °C for 18–24 h. According to CLSI guidelines, a strain is deemed to be ESBL-producing if there is at least a three-fold reduction in the MIC value of the antimicrobial agent when tested in combination with clavulanic acid.

### 4.5. Next-Generation Sequencing

The bacterial suspension underwent DNA extraction utilizing the Quick-DNA Fungal/Bacterial Miniprep Kit, D6005 (Zymo Research, Murphy Ave., Irvine, CA, USA) and following the prescribed protocol. Subsequently, paired-end reads derived from the DNA were identified using an Illumina NextSeq 500 sequencer [82]. The methodology utilized by Illumina products, including in this study, adopts a “pair end” approach. In this technique, single-stranded DNA strands are anchored with oligonucleotides during bridge amplification, while the complementary strand is inserted and bridged. Following this, the reverse strand is eliminated, and the fluorescently labeled linked nucleotides are read during the sequencing process [83,84].

Nucleotide sequences were determined through next-generation sequencing utilizing an Illumina NextSeq 500 sequencer (Illumina, located in San Diego, CA, USA), following established protocols [63]. For the reversible terminator sequencing (RTS) approach, the Illumina^®^ Nextera XT DNA Library Preparation Kit and the Nextera XT Index Kit v2 Set B (Illumina, San Diego, CA, USA) were employed to construct Illumina-specific libraries. The DNA samples were diluted to a final concentration of 0.2 ng/μL in nuclease-free water (Promega, Madison, WI, USA) with a total volume of 2.5 μL. The reaction components were utilized in reduced volumes. During the tagmentation reaction, 5 μL of Tagment DNA buffer and 2.5 μL of Amplicon Tagment Mix were combined. Tagmentation of the samples was carried out at 55 °C for 6 min using the GeneAmp PCR System 9700 (manufactured by Applied Biosystems/Thermo Fisher Scientific, located in Foster City, CA, USA). Subsequently, the samples were allowed to cool to 10 °C before adding 2.5 μL of neutralized tagment buffer, and neutralization was performed for 5 min at room temperature.

For the library amplification step, 7.5 μL of Nextera PCR Master Mix and 2.5 μL each of the i5 and i7 index primers were combined with the tagmented DNA samples. The index primers were incorporated into the library DNA via 12 PCR cycles, with each cycle consisting of the following stages: 95 °C for 10 s, 55 °C for 30 s, followed by 72 °C for 30 s. Following the PCR cycles, the samples were incubated at 72 °C for 5 min and then cooled to 10 °C. Subsequently, the libraries underwent purification utilizing the Gel/PCR DNA Fragments Extraction Kit from Geneaid Biotech Ltd. (Taipei, Taiwan). The concentration of the purified libraries was assessed, and the libraries were combined and denatured. The denatured library pool, with a final concentration of 1.8 pM, was loaded onto a NextSeq 500/550 High Output flow cell and subjected to sequencing using an Illumina^®^ NextSeq 500 sequencer (Illumina, San Diego, CA, USA).

### 4.6. Bioinformatic Analysis

Quality control of the raw sequences was carried out using FastQC v0.11.9 [85] and Fastp v0.23.2-3 [86]. Sequences of insufficient quality were filtered using TrimGalore v0.6.6. MEGAHIT v1.2.9 [87]. was employed to align read sequences into longer sequences (contigs). Quality control of the contigs was performed using QUAST v5.2 [38] and Busco v5 [88] Genome features were estimated using GenomeScope v2.2 [89]. Prodigal v2.6.3 [90]. was utilized to identify all possible open reading frames (ORFs) from the resulting contigs. ARGs among the ORFs were identified using the Resistance Gene Identifier (RGI) v5.1.0 against the CARD database [91]. Only genes meeting the STRICT threshold criteria defined by the CARD database, with at least 90% sequence identity and coverage, were considered.

To investigate the potential mobility of the identified resistance genes, we utilized MobileElementFinder (version 1.0.3) [92], a tool designed to predict mobile genetic elements (MGEs) on contigs. Only antibiotic resistance genes (ARGs) located within the vicinity of the longest E. coli-specific complex transposon in the database were deemed potentially mobile. PlasFlow version 1.1 [93] software and mlplasmids version 2.1 [94] software were employed to explore the plasmid origin of contigs, while the presence of phage genomes on contigs was assessed using VirSorter version 2.2.2 [95] software. Results pertaining to MGEs, plasmids, and phage genomes were filtered to include hits within 10,000 base pairs. For species identification, we utilized Checkm version 1.2.2 [96] software and Kraken version 1.1.1 [97] software. ResFinder version 4.1 [98,99,100] was employed to search for chromosomal point mutations, and Snippy version 4.6.0 was utilized to track polymorphisms in the genome. Ectyper version 1.0 was used for serotyping [101], and VirulenceFinder 2.0 was employed to monitor changes in virulence factors [99,102,103]. Genomic diversity analysis among genomes was performed using Average Nucleotide Identity (ANI) version 2.0 software to conduct taxonomic analysis of genomes from various phylogenetic lineages [104]. As for bioinformatic reference analysis, we utilized the *E. coli* (SYNB8802 strain) genome GCF_020995495.1 available in the National Center for Biotechnology Information (NCBI) database, which exhibited the closest full RefSeq genome overlap [105].

## 5. Conclusions

Our results raise particular concern regarding cephalosporins. They not only have outstanding significance and utilization in veterinary medicine but also play a crucial role in public health. It is worth noting the paradoxical possibility that certain antibiotic agents may increase the rate of acquiring antibiotic resistance [106]. Clarifying the relationships between specific cross-resistances holds paramount importance in choosing practical therapeutic strategies.

## Figures and Tables

**Figure 1 antibiotics-13-00247-f001:**
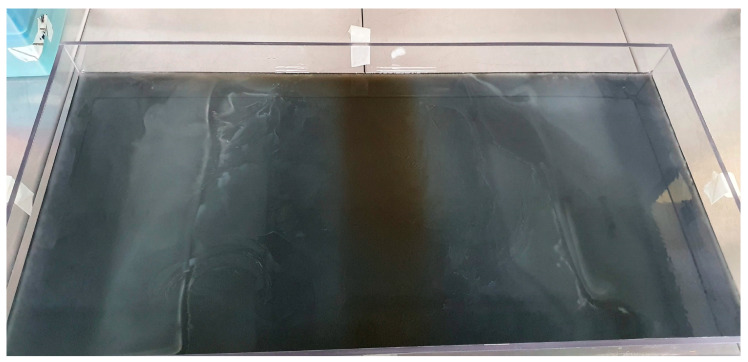
MEGA-plate overgrown with *E. coli* bacteria during 5 days of incubation against increasing amoxicillin concentrations.

**Figure 2 antibiotics-13-00247-f002:**
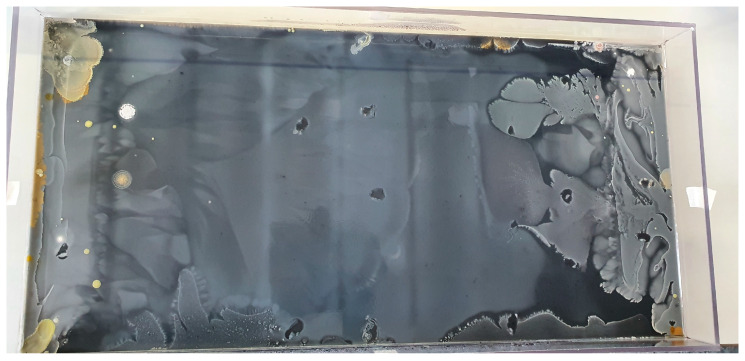
MEGA-plate overgrown with *E. coli* bacteria during 13 days of incubation against increasing cefotaxime concentrations. The tree-like growth patterns of individual branched strains are particularly well observable.

**Table 1 antibiotics-13-00247-t001:** Effects of increasing concentrations of amoxicillin on the MICs of the test compounds. Except for oxytetracycline and florfenicol, an increase in MIC was observed for all other drugs.

Sample	AMX	CTX	ENR	COL	OTC	PSA	FLO	NEO	CFR	CFT	CFQ
µg/mL											
0× AMX	4	0.03	0.003	0.5	2	8	16	16	0.25	0.06	0.06
1× AMX	**4**	**0.125**	0.003	0.5	2	8	16	16	0.25	**0.25**	**0.125**
10× AMX	**8**	**0.25**	**0.007**	0.5	2	8	16	16	0.25	**0.25**	**0.125**
100× AMX	**256**	**8**	**0.06**	**128**	2	**256**	16	**64**	**16**	**4**	**2**
1000× AMX	**256**	**8**	**0.06**	**512**	2	**256**	16	**64**	**16**	**4**	**8**

AMX—amoxicillin, CTX—cefotaxime, ENR—enrofloxacin, COL—colistin, OTC—oxytetracycline, PSA—potent sulphonamide (sulfamethoxazole, trimethoprim), FLO—florfenicol, NEO—neomycin, CFR—ceftriaxone, CFT—ceftiofur, and CFQ—cefquinome. Values in bold indicate an increase in amoxicillin-induced MIC.

**Table 2 antibiotics-13-00247-t002:** Effects of increasing concentrations of cefotaxime on the MICs of the test compounds. An inducing effect on MIC values was observed for all active ingredients in the case of cefotaxime application.

Sample	CTX	AMX	ENR	COL	FLO	OTC	PSA	NEO	CFR	CFT	CFQ
µg/mL											
0× CTX	0.03	8	0.003	0.5	16	2	8	16	0.25	0.06	0.06
1× CTX	**0.125**	8	0.003	0.5	16	2	8	16	0.25	0.06	0.06
10× CTX	**0.25**	**32**	0.03	0.5	**128**	**4**	**16**	16	0.25	0.06	0.06
100× CTX	**4**	**>512**	**0.125**	**8**	**256**	**4**	**32**	**64**	**16**	**8**	**4**
1000× CTX	**16**	**>512**	**0.125**	**32**	**256**	**16**	**64**	**128**	**64**	**32**	**8**

CTX—cefotaxime, AMX—amoxicillin, ENR—enrofloxacin, COL—colistin, FLO—florfenicol, OTC—oxytetracycline, PSA—potent sulphonamide (sulfamethoxazole, trimethoprim), NEO—neomycin, CFR—ceftriaxone, CFT—ceftiofur, and CFQ—cefquinome. Values in bold indicate an increase in cefotaxime-induced MIC.

**Table 3 antibiotics-13-00247-t003:** Results of the extended-spectrum beta-lactamase (ESBL) detection study with the CLSI-recommended combination of ceftazidime (CTZ) and cefotaxime (CTX) with clavulanic acid (CLA), regarding the amoxicillin samples.

Sample	CTZ	CTZ + CLA	Difference	CTX	CTX + CLA	Difference
	(µg/mL)			(µg/mL)		
0× AMX	0.03	0.03	0×	0.03	0.03	0×
1× AMX	0.06	0.03	2×	0.125	0.06	2×
10× AMX	0.25	0.125	2×	0.25	0.125	2×
100× AMX	64	32	2×	8	2	4×
1000× AMX	64	32	2×	8	2	4×

AMX—amoxicillin, CTZ—ceftazidime, CTZ + CLA—ceftazidime clavulanic acid, CTX—cefotaxime, and CTX + CLA—cefotaxime clavulanic acid.

**Table 4 antibiotics-13-00247-t004:** Results of the extended-spectrum beta-lactamase (ESBL) detection study with the CLSI-recommended combination of ceftazidime (CTZ) and cefotaxime (CTX) with clavulanic acid (CLA), regarding the cefotaxime samples.

Sample	CTZ	CTZ + CLA	Difference	CTX	CTX + CLA	Difference
	(µg/mL)			(µg/mL)		
0× CTX	0.03	0.03	1×	0.03	0.03	1×
1× CTX	0.03	0.03	1×	0.125	0.125	1×
10× CTX	0.06	0.06	1×	0.25	0.125	2×
100× CTX	8	4	2×	4	2	2×
1000× CTX	8	8	1×	16	8	2×

CTX—cefotaxime, CTZ—ceftazidime, CTZ + CLA—ceftazidime clavulanic acid, CTX—cefotaxime, and CTX + CLA—cefotaxime clavulanic acid.

**Table 5 antibiotics-13-00247-t005:** During the five-day research period, we recorded and classified all mutations in each sample relative to the mutation type and the reference strain utilized for analysis. Additionally, the number of mutations induced by the drug presence was compared to the untreated (0× AMX) sample and is denoted in parentheses with a plus sign.

Mutation Type		0× AMX	1× AMX	10× AMX	100× AMX	1000× AMX
Complex *	Identified	121	119	116	115	117 (+1)
	All	311	286	193	290	294
Deletion	Identified	20	20	20	19	20
	All	41	41	42	41	41
Insertion	Identified	4	4	4	3	5 (+1)
	All	14	15	15	13	15
SNP **	Identified	786	781	774	781 (+6)	789 (+5)
	All	1447	1364	1393	1381	1410

* A compound mutation that may involve multiple insertions, deletions, and substitutions; ** single-nucleotide polymorphism. AMX—amoxicillin.

**Table 6 antibiotics-13-00247-t006:** The cumulative count of mutations observed and identified in each sample, categorized by mutation type relative to the reference strain used for analysis over the 5 days of the study. The number of mutations observed, induced by the presence of the drug, is indicated in parentheses with a plus sign, compared to the untreated (0× other active ingredient) sample.

Mutation Type		0× CTX	1× CTX	10× CTX	100× CTX	1000× CTX
Complex *	Identified	116	120	118 (+1)	117 (+3)	116 (+2)
	All	311	309	327	297	305
Deletion	Identified	19	20	20	23 (+4)	22 (+3)
	All	38	37	38	42	41
Insertion	Identified	3	3	3	16 (+13)	14 (+11)
	All	12	12	13	42	33
SNP **	Identified	774	779 (+7)	784 (+13)	928 (+161)	927 (+174)
	All	1448	1486	1585	1641	1656

* A compound mutation that may involve multiple insertions, deletions and substitutions; ** single-nucleotide polymorphism. CTX—cefotaxime.

**Table 7 antibiotics-13-00247-t007:** Mutations affecting genes relevant for antimicrobial resistance in the presence of amoxicillin (at different concentrations, see columns 1–5) were observed as complex, insertion, or single-nucleotide polymorphisms (SNPs).

Gene	1	2	3	4	5	Nucleotide Acid Replacement	Effect	Product
COMPLEX								
*vgrG*					x	AGG-CGT	synonymous variant c.1228_1230delAGGinsCGT p.411	type VI secretion system tip protein
INSERTIO								
*ftsK*					x	A-T	frameshift variant c.2256_2257insT p. Gln753fs	DNA translocase
SNPs								
*ampC*				x	x	undefined	*ampC*-promoter n.-11C>T	undefined
*frdD*				x	x	G-A	missense variant c.353C>T p. Thr118Ile	fumarate reductase subunit
*kbaZ*				x		G-C	synonymous variant c.243G>C p. Pro81Pro	tagatose-bisphosphate aldolase subunit
*yhhZ*				x		C-A	missense variant c.528C>A p. Ser176Arg	Hcp1 family type VI secretion system effector
*ugpC*				x		G-C	synonymous variant c.261C>G p. Leu87Leu	sn-glycerol 3-phosphate ABC transporter ATP binding protein
*aceF*				x		T-C	synonymous variant c.387T>C p. Asp129Asp synonymous variant c.399T>C p. Ala133Ala	pyruvate dehydrogenase complex dihydrolipoyllysine-residue acetyltransferase
*ompN*				x		A-G	synonymous variant c.75T>C p. Tyr25Tyr	porin
*pta*					x	G-T	missense variant c.208C>A p. Pro70Thr	phosphate acetyltransferase
*tyrB*					x	A-G	synonymous variant c.426A>G p. Gly142Gly	aromatic amino acid transaminase
*acrB*					x	A-T	missense variant c.145T>A p. Tyr49Asn	efflux RND transporter permease
*vgrG*					x	C-T A-G T-G	missense variant c.1454C>T p. Thr485Ile synonymous variant c.1236A>G p. Ser412Ser synonymous variant c.1437T>G p. Gly479Gly	type VI secretion system tip protein

1—0× AMX; 2—1× AMX; 3—10× AMX; 4—100× AMX; 5—1000× AMX. AMX—amoxicillin. SNPs—single-nucleotide polymorphisms.

**Table 8 antibiotics-13-00247-t008:** Mutations affecting genes relevant for antimicrobial resistance were complex in the presence of cefotaxime (at different concentrations, see columns 1–5).

Gene	1	2	3	4	5	Nucleotide Acid Replacement	Effect	Product
*ampC*			x	x	x	GCC-CCA GTA-GAA CGG-GGG CGC-CAC	*ampC*-promoter p.A2P *ampC*-promoter p.V4E *ampC*-promoter p.R11G *ampC*-promoter p.R8H	A->P amino acid change V->E amino acid change R->G amino acid change R->H amino acid change
*ampC*				x	x	undefined	*ampC-*promoter n.32T>A	T->A amino acid change Phenotype amoxicillin, amoxicillin-clavulanic acid, ampicillin, ampicillin-clavulanic acid, cefixime, cefotaxime, cefoxitin, ceftazidime, piperacillin resistance because of beta-lactamase enzyme production.
*sstT*				x	x	T-GG	frameshift variant & missense variant c.1042delTinsGG p. Ser348fs	Serin/threonine transporter
*adk*			x	x		GAAAG-TAAAT GAAA-TAAT	missense variant c.420_424delGAAAGinsTAAAT p. Val142Leu missense variant c.420_423delGAAAinsTAAT p. Lys141Asn	adenylate kinase
*kbaY*				x		TCAT-CCAT	synonymous variant c.75_78delTCATinsCCAC p.27	tagatose-bisphosphate aldolase subunit
*fdoH*					x	CTGGAA-GGGCAT	missense variant c.76_81delTTCCAGinsATGCCC p. PheGln26MetPro	formate dehydrogenase O subunit beta

1—0× CTM; 2—1× CTM; 3—10× CTM; 4—100× CTM; 5—1000× CTM. CTM—cefotaxime. A-alanine; P-proline; V-valine; E-glutamic acid; R-arginine; G-glycine; H-histidine; T-threonine.

**Table 9 antibiotics-13-00247-t009:** Mutations affecting genes relevant for antimicrobial resistance were deletions, specifically under the influence of cefotaxime.

Gene	1	2	3	4	5	Nucleotide Acid Replacement	Effect	Product
*mutL*				x	x	AGCTGGC-A	disruptive inframe deletion c.215_220delTGGCGC p.Leu72_Ala73del	DNA mismatch repair endonuclease
*vat*				x	x	AC-A	frameshift variant c.1483delG p. Val495fs	vacuolating autotransporter toxin
*ybiO*				x	x	CT-C	frameshift variant c.138delA p. Ala47fs	mechanosensitive channel protein
*vceG*				x		GA-G	frameshift variant c.1013delA p. Asn338fs	cell division protein

1—0× CTM; 2—1× CTM; 3—10× CTM; 4—100× CTM; 5—1000× CTM. CTM—cefotaxime.

**Table 10 antibiotics-13-00247-t010:** Mutations affecting genes relevant for antimicrobial resistance were insertions, specifically under the influence of cefotaxime.

Gene	1	2	3	4	5	Nucleotide Acid Replacement	Effect	Product
*narQ*				x	x	C-CT	frameshift variant c.49dupT p. Tyr17fs	nitrate/nitrite two-component system sensor histidine kinase
*nfeF*				x	x	C-CG	frameshift variant c.16dupC p. Arg6fs	NADPH-dependent ferric chelate reductase
*rnpB*				x	x	T-TC	intragenic variant n.1764344_1764345insC	RNase P RNA component class A
*rsxD*				x	x	T-TC	frameshift variant c.274dupC p. Leu92fs	electron transport complex subunit
*ptsP*				x	x	T-TG	frameshift variant c.268dupC p. His90fs	phosphoenolpyruvate--protein phosphotransferase
*ubiC*				x	x	T-TG	frameshift variant c.420dupG p. Arg141fs	chorismate lyase
*msrA*				x	x	T-TG	frameshift variant c.367dupC p. Gln123fs	peptide-methionine (S)-S-oxide reductase
*ytfR*				x	x	C-CA	frameshift variant c.1230dupA p. Val411fs	sugar ABC transporter ATP-binding protein
*gudP*					x	T-TC	frameshift variant c.927dupG p. Ile310fs	galactarate/glucarate/glycerate transporter
*yicI*					x	C-CG	frameshift variant c.1065dupC p. Val356fs	alpha-xylosidase
*maeA*					x	G-GT	frameshift variant c.21dupA p. Gln8fs	malate dehydrogenase
*xapB*				x		C-CT	frameshift variant c.564dupA p. Ala189fs	xanthosine/proton symporter
*hypF*				x		G-GC	frameshift variant c.1544dupG p. Glu516fs	carbamoyl transferase
*phoP*				x		A-AT	frameshift variant c.601dupA p. Ile201fs	two-component system response regulator
*tssI*				x		A-AG	frameshift variant c.1976dupG p. Val660fs	type VI secretion system tip protein TssI/VgrG
*marR*				x		C-CA	frameshift variant c.377dupA p. Asn126fs	multiple antibiotic resistance transcriptional regulator

1—0× CTM; 2—1× CTM; 3—10× CTM; 4—100× CTM; 5—1000× CTM. CTM—cefotaxime.

**Table 11 antibiotics-13-00247-t011:** Mutations affecting genes relevant for antimicrobial resistance were SNPs, specifically under the influence of cefotaxime.

Gene	1	2	3	4	5	Nucleotide Acid Replacement	Effect	Product
*acrR*			x	x	x	T-C	missense variant c.458T>C p. Met153Thr	multidrug efflux transporter transcriptional repressor
*ampC*				x	x	C-T	missense variant c.922G>A p. Ala308Thr	cephalosporin-hydrolyzing class C beta-lactamase EC-5
*acrB*			x	x		G-C C-T	missense variant c.1693C>G p. Pro565Ala missense variant c.2906G>A p. Arg969Gln	efflux RND transporter permease
*robA*			x	x		C-T	missense variant c.467G>A p. Arg156His synonymous variant c.849G>A p. Leu283Leu	MDR efflux pump *acrAB* transcriptional activator
*mrcA*				x	x	G-A	missense variant c.772G>A p. Ala258Thr	peptidoglycan glycosyltransferase/peptidoglycan DD-transpeptidase
*mdtB*				x		G-A	missense variant c.805G>A p. Ala269Thr	multidrug efflux RND transporter permease subunit
*emrR*					x	T-C	missense variant c.478T>C p. Ser160Pro	multidrug efflux transporter *emrAB* transcriptional repressor

1—0× CTM; 2—1× CTM; 3—10× CTM; 4—100× CTM; 5—1000× CTM. CTM—cefotaxime.

## Data Availability

The datasets used and/or analyzed during the current study are available from the corresponding author on reasonable request. The sequencing files are available at the https://www.ncbi.nlm.nih.gov/bioproject/prjna1064554, accessed on 14 January 2024.

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
