# Peer review of "In Vitro Microevolution and Co-Selection Assessment of Amoxicillin and Cefotaxime Impact on Escherichia coli Resistance Development"

_antibiotics, 2024, doi:10.3390/antibiotics13030247_

Round 1

Reviewer 1 Report

Comments and Suggestions for Authors

The most interesting observation of the study was that in the presence of amoxicillin resistance, the MICs of oxytetracycline and florfenicol remain unchanged, which make them potential chemotherapeutic agents for treatment. It seems like a missed opportunity to not have further explored this mechanism further. The methods describe that for the MIC assays 'we designated 0.25× of the initial MIC value of florfenicol (16 μg/mL) as the 1× concentration.' The authors do not provide any rationale for this, and it is hard to understand why this was selected as is. The SNPs and other mutation that were identified as presented in Tables 7-11 are wholly expected and do not seem to expand our general understanding of antimicrobial resistance mechanisms in any way. However, in the context of the study this is sufficient and overall, the data presented in this manuscript is straightforward and easy to understand.

Author Response

Dear Reviewer 1,

Thank you very much for your comments, our response is attached.

Yours sincerely,
Adam Kerek

Reviewer 2 Report

Comments and Suggestions for Authors

The manuscript byÁdám Kerek, et al., “In Vitro Microevolution and Co-selection Assessment of Amox[1]icillin and Cefotaxime Impact on Escherichia coli Resistance Development” explored an interesting and important question. The manuscript studied the development of phenotypic and genotypic resistance to beta-lactam antibiotics, focusing on amoxicillin and cefotaxime. The research method is rigorous and the discussion is sufficient. The results are all well presented. I think the manuscript is worth publishing.

Author Response

Dear Reviewer 2,

Thank you very much for your comments, our response is attached.

Yours sincerely,
Adam Kerek

Reviewer 3 Report

Comments and Suggestions for Authors

After reading the content of the original article entitled "In Vitro Microevolution and Co-selection Assessment of Amoxicillin and Cefotaxime Impact on Escherichia coli Resistance Development", I believe that the data obtained are very interesting and correctly presented. Below I would like to present short list of improvements:

- page 2: In the Introduction, it is worth discussing in one or two sentences what MEGA-plate is and what its advantages are.

- The discussion lacks the highlight of one important clinical implication, which results directly from the obtained results. Even the use of high concentrations of antibiotics (greater than or equal to the MIC) may generate antibiotic resistance and genotypic changes in pathogens over time. It is worth distinguishing this in some way, because most often such conclusions are formulated in the context of sub-MIC values and ineffective treatments.

Author Response

Dear Reviewer 3,

Thank you very much for your comments, our response is attached.

Yours sincerely,
Adam Kerek

Reviewer 4 Report

Comments and Suggestions for Authors

In this manuscript, Kerek et al. provide evidence of adaption of bacteria to increased concentration of antimicrobial agents, in specific amoxicillin and cefotaxime.  Also, this adaption confers resistance to other antimicrobial agents. The genetic basis of this phenotypic observation was studied based on NGS analyses and specific mutations were recorded.  MEGA-plate method was used for generation of mutants.

The results are very interesting and conclusions are supported by extracted data.

I could propose mRNA analyses, too, in order to assess  expression of studied genes.

Author Response

Dear Reviewer 4,

Thank you very much for your comments, our response is attached.

Yours sincerely,
Adam Kerek

Round 2

Reviewer 1 Report

Comments and Suggestions for Authors

The authors have addressed the comments and the revisions highlighted in the current version of the manuscript is satisfactory.

Reviewer 3 Report

Comments and Suggestions for Authors

Thank you for following and applying all the suggested comments. I believe that the manuscript is suitable for publication.